# An affordable solution for investigating zebra finch intracranial electroencephalography (iEEG) signals

**Mohammad-Mahdi Abolghasemi[1], Shahriar Rezghi Shirsavar[1], Amirreza Bahramani[1,2], Milad Yekani**●[1]*

**1** School of Cognitive Sciences, Institute for Research in Fundamental Sciences (IPM), Tehran, Iran,
**2** Department of Electrical Engineering, Sharif University of Technology, Tehran, Iran

* miladyekani@ipm.ir

## Abstract

The zebra finch is a well-studied animal model for investigating the neural mechanisms of vocal learning, and electrophysiology is the primary technique for understanding their song system. Most of the studies on zebra finches have focused on intracerebral recordings. However, these methods are only affordable for limited laboratories. Recently, different open-source hardware for acquiring electroencephalography (EEG) signals has been developed. It's unclear whether these solutions suit zebra finch studies as they have not been evaluated. Electrocorticography signals can provide a preliminary guide for more in-depth inquiries and also aid in understanding the global behavior of the bird's brain, as opposed to the more common localized approach. We present a detailed protocol for acquiring intracranial electroencephalography (iEEG) data from zebra finches using the OpenBCI Cyton board, an open-source device. We implemented stainless steel electrodes on the brain's surface and recorded the brain signals from two recording sites above two auditory-responsive nuclei. To validate our method, we ran two different experiments. In the first experiment, we recorded neural activity under various concentrations of isoflurane and extracted the suppression duration to measure anesthesia depth. In the second experiment, we head-fixed the birds and, under light anesthesia, presented them with various auditory stimuli to evaluate event-related potentials (ERP) and generate spectrograms. The results showed a significant increase in suppression duration with deeper anesthesia, and the ERP and spectrogram responses to auditory stimuli differed accordingly. These findings indicate that using our methodology, one can successfully collect iEEG signals from zebra finches. These findings pave the way for future studies to use iEEG to investigate bird cognition in a more affordable way.

Data availability statement: https://github.com/finch-zebra/ieeg

Funding: The author(s) received no specific funding for this work.

Competing interests: The authors have declared that no competing interests exist.

## Introduction

The investigation of the avian neural system provides a promising avenue for illuminating the underlying neural mechanisms of vocal learning and even speech acquisition in humans [1]. As an exceptional species that can acquire singing via vocal imitation from parents, humans exhibit noteworthy similarities with songbirds regarding complex sound production and learning. Thus, songbirds have emerged as effective models for neuroscientific inquiries concerning vocal learning [2]. A range of research methods is used in the extensive investigations of the bird's vocal system, including electrophysiology [3], histology [4], genetics [5,6], and behavioral studies [7]. The bulk of current knowledge concerning the avian vocal system has been derived from experimental studies conducted on the zebra finch (*Taeniopygia guttata*), a widely accepted avian laboratory model species [8]. Invasive neural recordings in zebra finches have revealed valuable insights into the song system and mechanisms of vocal learning. The ultimate goal in this line of research is to understand the neural mechanism of creating complicated motor sequences and vocal production and translate these findings to the human brain.

Human brain studies are restricted by a limited capacity for intervention due to ethical and technical considerations. These limitations force human research to use noninvasive methods like electroencephalography (EEG). Nevertheless, testing these methods on animal models provides valuable insights, establishing a robust foundation for verifying their applicability. These signals are clinically relevant to consciousness alterations, seizure activity, anesthesia, and sleep states. As an alternative approach, since animals enable invasive practices, studies utilizing the intracranial version of EEG (iEEG) have provided more valuable insight into the nature of acquired signals than EEG due to a higher signal-to-noise ratio [9]. Animal iEEG studies also have an extensive history that provides complementary data to human research. Specifically, animal iEEG models exploring pharmacological agents' impact on the brain represent a crucial translational inquiry area [10].

IEEG recording has been conducted to study sleep and migratory behaviors in birds [11]. Research on birds' sleep patterns, which are essential from an evolutionary perspective, has also been conducted using iEEG. Furthermore, brain activity related to memory consolidation during the critical period of song learning in juvenile finches can be detected by this method [12]. The research literature on working with birds is full of extracellular recording studies, in which we can examine the behavior of a neuron during the production or hearing of a song with a very high accuracy. Despite the importance of animal iEEG, avian iEEG has been studied very limited compared to other animals.

IEEG studies in birds have great translational potential. In this concept, reproducing bird songs from iEEG signals can be a suitable model for the brain-computer interface of speech. The advantage of this method is its ease of implementation, which allows it to be performed before other neuronal recording experiments and during behavioral studies. Consequently, the higher cost and energy of conducting neural recording experiments, which make them less feasible, will be evaded.

With the increasing demand for neural data, scientists seek more accessible ways to democratize scientific research [13]. To this end, many entities produce tools, programs, and open-source data sets. Given the importance of EEG studies, open-source hardware and software solutions are being introduced in this field. OpenBCI offers a versatile open-source hardware platform for various biosensing applications, including EEG, EMG, and ECG signals [14]. At its core, the OpenBCI Cyton board integrates several key components: a PIC32MX250F128B microcontroller, a ChipKIT UDB32-MX2-DIP bootloader, an ADS1299 analog-to-digital converter with eight channels (expandable to 16) and a maximum sampling rate of 16 kHz, and a 3-axis LIS3DH accelerometer. Users can configure EEG channels in either monopolar or bipolar modes, while the board also accommodates five external digital inputs and three analog inputs. Moreover, the device supports data streaming over Wi-Fi via Transmission Control Protocol (TCP) [15].

We conducted two experiments to test the efficacy of our method. In the first experiment, we recorded signals under different concentrations of isoflurane. The suppression duration was used as a proxy for anesthesia depth. In the second experiment, following the recovery and adaptation period, we recorded iEEG from two electrodes above two auditory-responsive nuclei while the birds were under light anesthesia. We presented various auditory stimuli, including the bird's own song (BOS) and its reversed version. Analysis of the recorded data revealed robust neural responses to all stimuli. Notably, the neural response patterns to BOS were distinct from those elicited by the other stimuli, supporting the validity of our method. Our study highlights the usefulness of OpenBCI in recording the iEEG activity of a bird's brain, particularly the zebra finch, for investigating auditory responses.

## Methods

The protocol described in this peer-reviewed article is published on protocols.io, https://dx.doi.org/10.17504/protocols.io.q26g7y2w8gwz/v3, and is included for printing as supporting information file 1 with this article.

### Animal subjects

We used five male zebra finches (> 60 days old) for experiment one and two male zebra finches for experiment two. All animals were obtained from the IPM birds garden aviary and kept in a local animal house after surgery. All procedures were approved by the Research Ethics Committee of the School of Cognitive Sciences (SCS) of the Institute for Research in Fundamental Sciences (IPM, protocol number 1402/40/1/2841).

### Electrodes and head-post implantation

Animals were kept under restricted food and water conditions before surgery. Anesthesia was induced using 2% isoflurane, and the animal was secured into the stereotaxic device after induction. We used our designed 3D-printed anesthesia mask compatible with the stereotaxic device to sustain gas delivery to the head-fixed bird. The head feathers were removed, and the incision site was prepared for surgery. After exposing the skull and identifying the bifurcation landmark, we determined the coordinates for the craniotomy sites relative to this bifurcation. Our goal was to position electrodes on two auditory-responsive nuclei: HVC (proper name) and the caudomedial mesopallium (CMM). The HVC site was located 0.2 mm anterior and 2.3 mm lateral, while the CMM site was positioned 2 mm anterior and 0.4 mm lateral from the bifurcation. Craniotomy sites were drilled, and after placing the electrodes, the skull was prepared for cement adhesion, and after applying the cement to the area, the head post was installed. The cement was allowed to be cured completely before proceeding. Detailed instructions can be found in the protocol section.

### Experimental procedures

**Experiment 1: Suppression duration recording under anesthesia.** We collected iEEG data from two recording electrodes implanted beneath the skull of the zebra finches during the different depths of anesthesia with isoflurane. After the surgery and electrode implantation, the OpenBCI recording was set up, and the signals were recorded with a 250 Hz

sampling rate for about 2 minutes in each of 2.5%, 2%, 1.5%, 1%, 0.6%, and 0.4% of isoflurane (Fig 1A). Before analysis, the data were preprocessed to ensure data quality and reliability. Standard preprocessing steps were employed, including a high-pass filter at 1 Hz, a low-pass filter at 80 Hz, and a notch filter at 50 Hz to remove noise, and the signal was re-referenced during recording by placing electrodes on the cerebellum. After that, the data was extracted based on the different levels of isoflurane for further analysis in suppression detections.

**Experiment 2: Auditory response recording under light anesthesia.** To record neural activity from HVC and CMM regions, five auditory stimuli were each presented 60 times with silent intervals randomly ranging from 2 to 4 seconds between them. The stimuli included BOS, reversed BOS, conspecific song, a 5 kHz tone, and white noise (Fig 1B). A highly synchronized trigger signal was essential during recording to accurately annotate each trial. Audio playback and trigger output to the OpenBCI device were managed using a Raspberry Pi 3. An optocoupler circuit was employed to isolate the OpenBCI device from the Raspberry Pi, preventing noise induction from the trigger signal. The task was programmed in C++ and optimized to achieve a latency of 1 ms, ensuring the required accuracy.

## Analysis

### Experiment 1: Suppression duration

Bursts are characterized by short episodes of partial reactivity or temporary spikes in neural firing, followed by a significant decrease or suppression of neural activity. We measured suppression duration in zebra finches under varying levels of isoflurane anesthesia. Suppression duration is a crucial metric for evaluating brain activity and detecting instances of burst suppression, which may indicate fluctuations in brain function or anesthesia-induced alterations in neural activity. Our objective was to explore the neural dynamics and anesthesia effects on zebra finches while administering different levels of isoflurane. The previous method for defining burst suppression relied on setting a fixed threshold for EEG voltage (typically between 0.5 and 20 μV) and classifying suppressions as segments where the voltage remained below this threshold for at least 0.5 seconds. This approach was problematic due to variations in amplitude and spectral characteristics across different pathological conditions and anesthetic effects. In 2013, Westover et al. introduced a new method involving an empirical approach, in which two experienced electroencephalographers manually segmented EEG data based on visual pattern recognition and clinical judgment. Additionally, the study developed an automatic segmentation method using

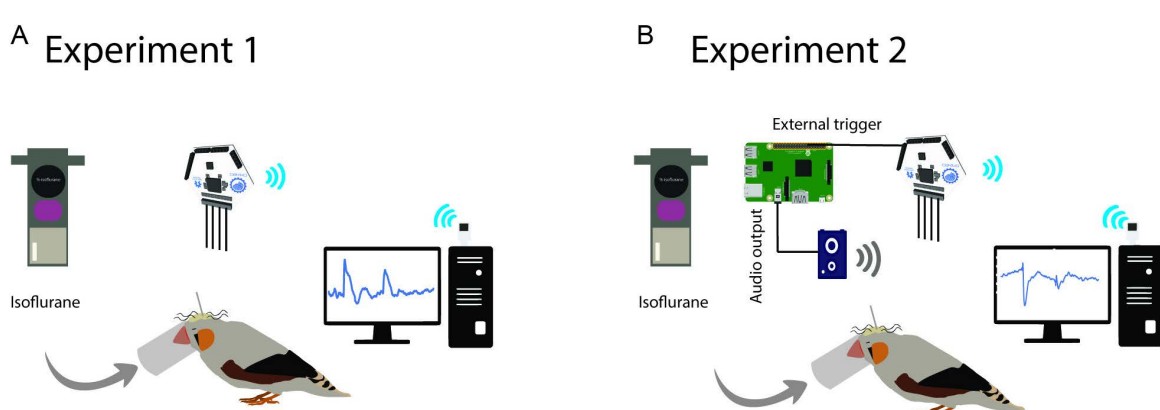

**Fig 1. Graphical abstract of two experiments.** Implanted parts on the bird's skull have been identified in different colors. The cream area on the skull shows the cemented region on top of the head. As demonstrated, all other compartments are fixed by cement on the skull. The curly wires represent recording and ground electrodes, and the perpendicularly attached bar (gray) indicates the head post. The electrodes are connected to the Cyton recording board (for a detailed explanation of connections, see the protocol.io). (A) In the first experiment, varying concentrations of isoflurane were used to achieve different depths of anesthesia. (B) In the second experiment, the finch was head-fixed under light anesthesia, and intracranial EEG (iEEG) data were recorded from implanted electrodes during an auditory task.

recursive variance estimation, which classifies bursts and suppressions based on the adaptive mean and variance of the EEG signal, optimized to match expert consensus. As a result, a real-time algorithm was created to automatically differentiate suppressions from non-suppressions (bursts) in EEGs. Using this algorithm, we rely on the total suppression duration for each task condition and compare these durations statistically. At each level of isoflurane (low, medium, and high), we compute the suppression duration for each 2-minute segment of the session and perform statistical analyses to compare the conditions [16].

To perform further statistical analyses, we sorted the dataset based on the isoflurane concentration into three distinct categories: low, medium, and high levels of anesthesia. Specifically, isoflurane below one percent was classified as low-level anesthesia, those above two indicated a high level, and values falling between 1 and 2 were classified as medium levels of anesthesia. To compare suppression durations across the three experimental conditions, a Friedman test was conducted as a nonparametric alternative for repeated measures. When significant differences were found, post hoc pairwise comparisons were performed using Wilcoxon signed-rank tests with Bonferroni correction to adjust for multiple comparisons.

### Experiment 2: Spectrogram and Event-Related Potential (ERP)

The data collected from OpenBCI was saved as.txt files. After importing the data, we applied a bandpass filter between 1 and 40 Hz to focus on the frequency range of interest and reduce noise. We then segmented the continuous data into individual trials based on trigger times corresponding to stimulus presentations. For the event-related potential (ERP) analysis, we performed baseline correction by subtracting the mean activity during the 500 ms preceding the stimulus presentation from each trial. This normalization step ensures that the post-stimulus activity is relative to the pre-stimulus baseline. We then averaged the baseline-corrected trials for each stimulus type to obtain the ERP waveforms. For the time-frequency analysis, we computed spectrograms using the Stockwell Transform [17,18], which provides a detailed time-frequency representation of the signal. To normalize each trial's spectrogram, we calculated the mean power over time and used it to adjust the trial's response, effectively controlling for inter-trial variability in power. After normalization, we averaged the spectrograms by stimulus type to produce the final time-frequency plots. All analyses were conducted using Python with the MNE 1.8 toolbox. To ensure the reliability of our results, we also replicated the analyses using custom scripts in MATLAB R2023b.

## Results

### Anesthetized recording

Our analysis revealed distinct changes in suppression duration across different stages of isoflurane anesthesia. As isoflurane was administered, we observed a gradual increase in the suppression duration, indicating an increase in burst suppression events (Fig 2C) and reduced brain activity, as evidenced by computed fast Fourier transforms in each of low, medium and high anesthesia for both anterior and posterior electrodes (Fig 2A). Additionally, we statistically compared the suppression durations across nine blocks of 120-second recordings at low, medium, and high anesthesia levels. The analysis of suppression durations across the three experimental conditions (low, medium, and high isoflurane) was conducted using a Friedman test. Significant differences were identified among the conditions for both electrode sites. The p-value for the Friedman test in the anterior electrode was 0.00841, while for the posterior electrode, it was 0.0015. Post hoc pairwise comparisons were performed using Wilcoxon signed-rank tests with Bonferroni correction to account for multiple comparisons. For the anterior electrode, the adjusted p-values for the pairwise comparisons were as follows: high vs. medium: 0.75, low vs. medium: 0.16, and low vs. high: 0.02 (Fig 2 B-top). For the posterior electrode, the adjusted p-values for the pairwise comparisons were 0.93, 0.023, and 0.023, respectively (Fig 2B-bottom). Moreover, to confirm whether the bird's responses were affected by the level of anesthesia, We compute and report the minimum and maximum duration, in addition to the average and standard error, to ensure that inter-individual variability is not substantial (Table 1).

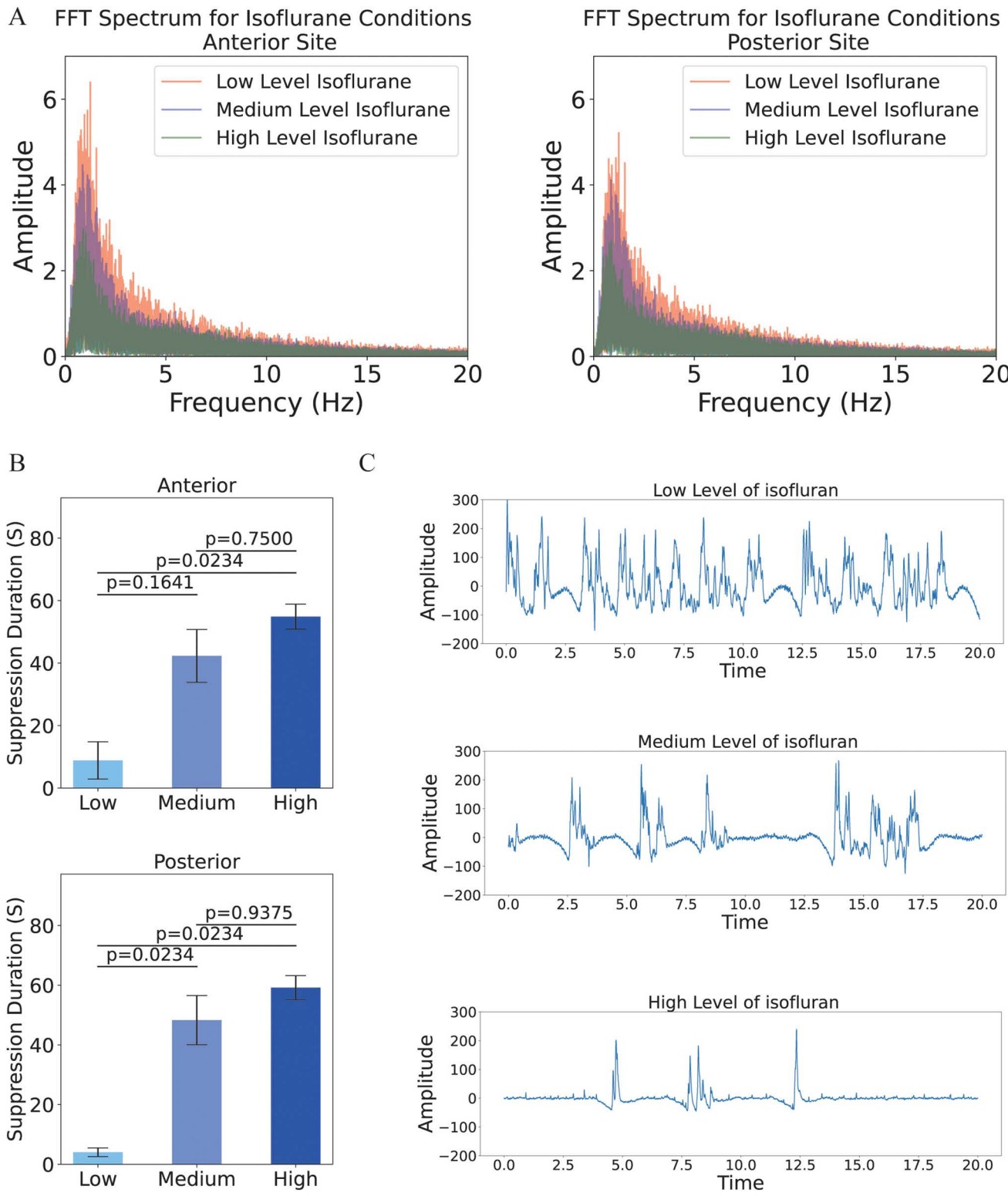

**Fig 2. FFT analysis of isoflurane levels, suppression duration comparison across conditions, and sample oscillations at different isoflurane levels.** (A) FFT analysis under low, medium, and high levels of isoflurane anesthesia, showing a decrease in power across various frequency ranges as isoflurane levels increase, with observations at both posterior (Right) and frontal (Left) sites. (B) Statistical pairwise comparisons of three different

isoflurane levels, with Bonferroni-adjusted p-values for anterior (top) and posterior (bottom) sites. (C) A 20-second frame of raw signal oscillations under low (top), medium (middle), and high (bottom) isoflurane anesthesia, illustrating a higher amplitude in the frequency range for low-level isoflurane.

**Table 1. Summary of duration measures, including minimum, maximum, average, and standard error.**

| Region | Condition | Mean | Min | Max | SE |
|---|---|---|---|---|---|
| Anterior | Condition 1 (Low) | 8.83 | 0.00 | 58.70 | 5.96 |
| Anterior | Condition 2 (Medium) | 42.32 | 16.29 | 80.00 | 8.45 |
| Anterior | Condition 3 (High) | 54.87 | 37.28 | 77.14 | 4.03 |
| Posterior | Condition 1 (Low) | 4.03 | 0.00 | 12.05 | 1.47 |
| Posterior | Condition 2 (Medium) | 48.30 | 21.06 | 80.00 | 8.24 |
| Posterior | Condition 3 (High) | 59.18 | 43.34 | 76.17 | 4.04 |

### Auditory response recording.

All five auditory stimuli elicited neural responses, evident in the ERP recordings from both electrodes (Fig 3A, B). Notably, there is a sharp response occurring approximately 200 ms after the stimulus presentation. This delay is expected, as the neural activity is primarily generated by the underlying nuclei—namely, HVC and the CMM—which are both responsive to higher-level auditory stimuli. Interestingly, the ERP response for the BOS differs significantly from the other stimuli; it exhibits a later rise with a smaller slope. This may be due to the distinct responses of these nuclei to the BOS. These differences in response were also distinctly observed in the HVC and CMM. A larger amplitude in the CMM compared to the HVC may result from the dominant role of the CMM in processing auditory stimuli, as is evident across all conditions (Fig 3A, B).

Fig 4A, B, shows the spectrogram of the neural responses. There is a rapid, short response in the Theta and Alpha frequency bands for all stimuli. These early responses can be attributed to neural activity generated by the primary auditory areas. Moreover, the neural response for BOS is again distinct, displaying more pronounced and sustained activity in the Delta band. An interesting phenomenon is that the increase in Delta band power is followed by an immediate decrease which can be an indication of the underlying neural dynamics.

Differences are also evident in the HVC and CMM. Specifically, the CMM shows a greater response to conspecific and Reverse BOS stimuli than BOS alone. However, the HVC response to the BOS is more prominent than the reverse BOS. Interestingly, the response to the BOS in HVC is also greater than this response in the CMM. Overall, both areas respond more strongly to song-like stimuli than to white noise or the 5 kHz tone, as seen in the spectrogram. For better illustration, the spectrograms of the presented audio stimuli are shown in Figs 4C and 3C.

## Discussion

Songbirds are the primary animal models for studying vocal perception, production, and learning [19]. In the past decades, zebra finch has been utilized as a significant model in these studies, and spike recording was the primary method in these experiments. However, the techniques used to track the activity of a single neuron are expensive [20] and complicated and provide a narrow field of view to capture the brain's overall activity [21]. Recently, there has been a growing interest in studying the cognitive processes in birds as they show fascinating abilities in doing high-level mental functions [22]. These functions are processed in a distributed manner in the brain, and to study them, we need to have a larger field of view. Our experiment tries to address the problems mentioned above. In summary, we attempt to solve two issues. First, to reduce the cost of the avian studies by using open-source solutions, and second,

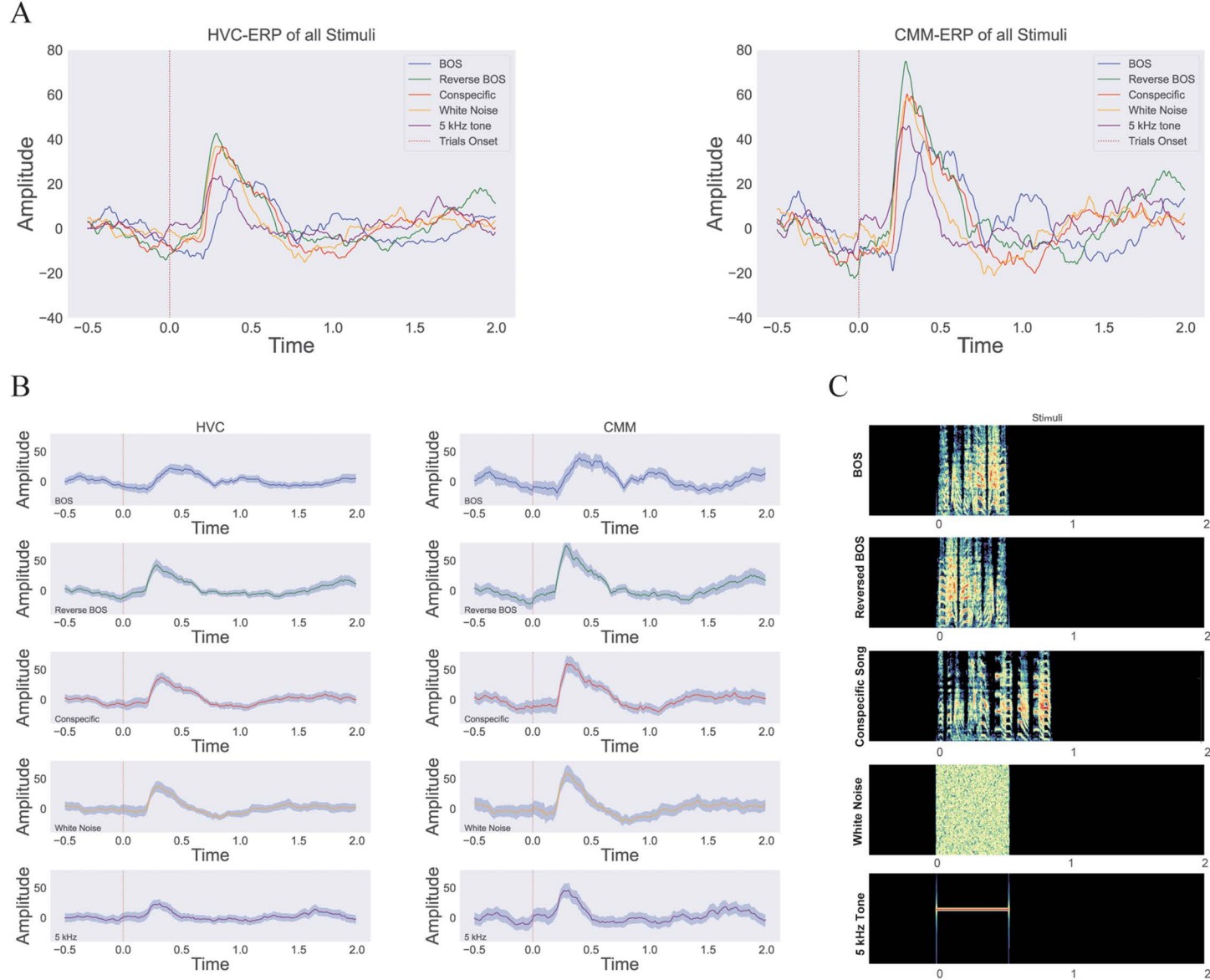

**Fig 3. Event-Related Potentials (ERPs) for all stimuli gathered, separated, and spectrogram of presented stimuli.** (A) Displays the ERPs of all presented auditory stimuli for HVC (Right) and CMM (Left). (B) Shows the separated ERPs of stimuli in HVC (Right) and CMM (Left). (C) Presents the spectrogram of the audio stimuli during the task.

to capture the brain's dynamic state during mental activity by recording iEEG. This paper presents a detailed protocol for recording iEEG from an anesthetized zebra finch. Previous experiments on birds' EEG under isoflurane anesthesia showed similar signal patterns to those of mammals [23]. This pattern is entirely detectable by visually inspecting a bird's EEG signal under surgical anesthesia, characterized by brief periods of burst activity and silent suppressed intervals between them [24]. We used suppression duration as an indicator of the anesthesia's depth in our experiment to test the efficiency of our method. The results show that this index is significantly higher in the high concentrations of gas than the low concentrations, indicating that our method successfully captures the brain's dynamic. Although

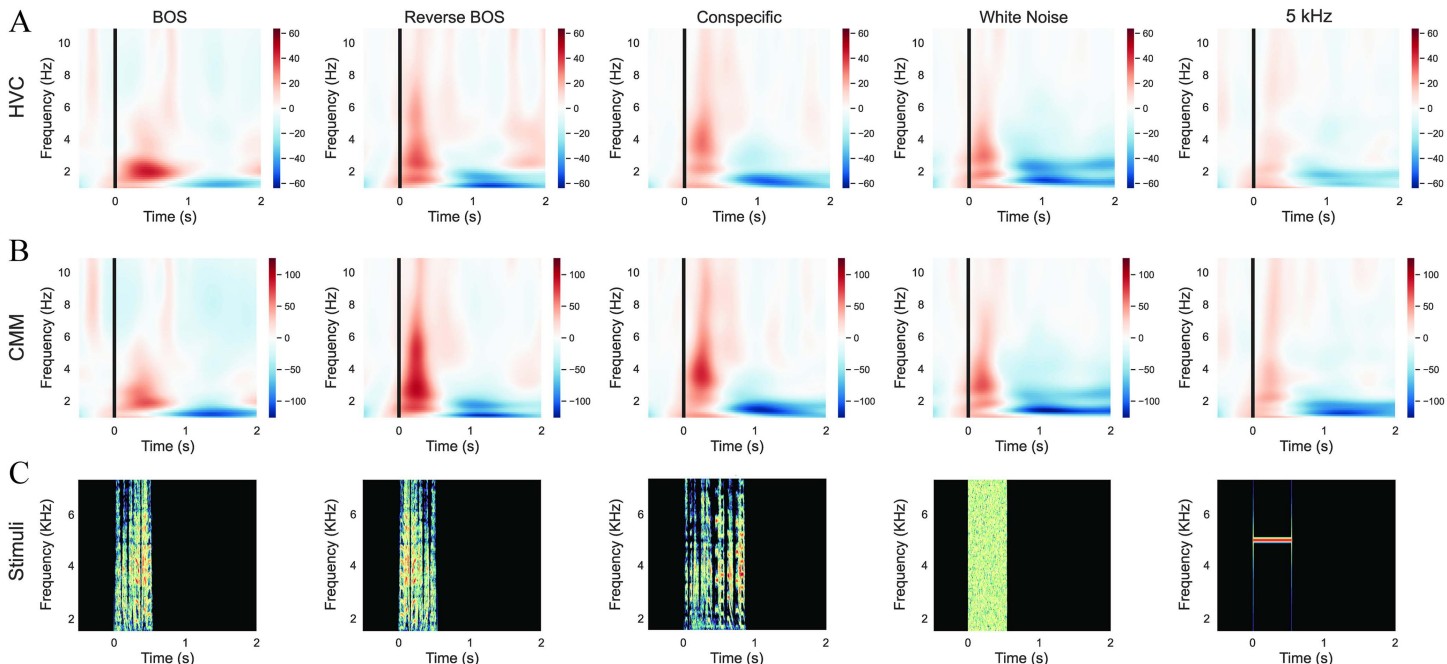

**Fig 4. Spectrograms for each stimulus condition in HVC, CMM, and audio stimuli.** (A) Spectrogram of HVC responses to the five stimuli: BOS, Reverse BOS, Conspecific, White Noise, and 5 kHz Tone. A distinct, selective response to BOS is observed in the HVC site. (B) Spectrogram of auditory stimuli responses in the CMM area, showing a greater response to Reverse BOS and Conspecific compared to the HVC area. (C) Spectrogram of the presented audio stimuli during the task.

we chose this experiment to show the validity of our method as Anesthesia produces a characteristic feature of brain dynamics that has been previously observed, there are some scientifically valuable facts in showing this phenomenon. The emergence of burst suppression in this species under anesthesia shows an evolutionarily conserved arousal mechanism. These findings can open future studies to understand a shared mechanism behind producing a conscious state in the brain. Comparative Studies related to conscious states in other animals can create insight into the serious questions about the evolutionary aspect of consciousness [25], and studies on birds' cognition regarding their high cognitive ability are in this category. Our ultimate aim in developing this method is to use it in studies related to the cognitive process in birds, particularly those connected to vocal perception and production. We conducted a second experiment with male zebra finches under light anesthesia, and they were presented with five different stimuli. Our results demonstrated an ERP response to all five stimuli, which were composed of two parts: an early activity in higher frequency bands, which probably originated from field L. This was followed by an increase in the power of the Delta band, which was later and specifically different for BOS. We think that this second phase is mostly generated by higher areas, especially HVC and CMM, in which the electrodes were implanted above them. Moreover, these responses in the Delta band were followed by a decrease in power, which can be the subject of further analysis of circuit-level mechanisms. Altogether, these observations can validate our method and showcase its use for investigating the neural correlates of vocal perception in an affordable way.

Although we observed notable differences between the responses to BOS and reverse BOS in the HVC and CMM electrodes, we should consider that localization can be improved by using higher-impedance electrodes and limiting the contact area. These modifications could also allow us to implant electrodes directly within the nuclei, enabling us to capture LFP signals with greater spatial resolution.

Our method offers a way to record chronic data from animals, as they can carry implanted materials for their lives. This recording approach is helpful when we want to capture the changes in brain function during aging and maturation (e.g., changes in vocal perception during the transition from the juvenile stage to adulthood) [26]. Given that the main noninvasive current method in BCI uses EEG signals to control effectors in humans, we think the zebra finch iEEG can be an appropriate model for the vocal prosthesis. Previous studies that have utilized the biomechanical model of the birds' vocal organs and fed them by neural signals gained tremendous success [27]. Future studies can improve this method fed those models and create a strong basic model for vocal prosthesis controlled by EEG signals. One of the early and valuable uses of EEG signals is detecting sleep states and disorders [28,29]. Low et al. have studied sleep in zebra finches and found mammalian-like patterns of brain activity in them during sleep [30]. The effect of sleep on developing song memory has been studied previously, and this phenomenon reveals an evolutionary fundamental rule of sleep in memory consolidation [31,32]. Our method can be utilized for studying sleep in zebra finches. The chronic recording ability of this method makes it suitable for studying the rule of sleep on song memory during development and the long-term effect of sleep deprivation in adults. Finally, the wireless communication protocol employed by OpenBCI presents an exciting opportunity for developing small-scale devices capable of chronic recording in freely behaving animals, especially in flying birds. Thanks to its open-source nature, research groups can modify these devices to improve their reliability for animal studies. While OpenBCI has proven useful in human studies, its application in animal studies has been limited until now. This study paves the way for future research endeavors to leverage this technology to collect a broader range of signals, especially local field potential. Birds are increasingly used in neuroscience and cognitive studies; however, the EEG signal has not been used extensively in this line of research. Our findings can trigger future studies to use iEEG in bird cognitive studies.

## Supporting information

**S1 File. Protocol for intracranial EEG recording in zebra finches.** Comprehensive experimental protocol describing materials, hardware configuration, electrode preparation, surgical procedure, data acquisition, auditory stimulus presentation, data analysis, post-operative care, and troubleshooting, as provided in the uploaded protocols.io document. (PDF)

## Author contributions

**Conceptualization:** Milad Yekani.

**Data curation:** Mohammad-Mahdi Abolghasemi, Shahriar Rezghi Shirsavar, Amirreza Bahramani, Milad Yekani.

**Formal analysis:** Mohammad-Mahdi Abolghasemi, Amirreza Bahramani.

**Investigation:** Mohammad-Mahdi Abolghasemi, Amirreza Bahramani, Milad Yekani.

**Methodology:** Mohammad-Mahdi Abolghasemi, Amirreza Bahramani, Milad Yekani.

**Project administration:** Milad Yekani.

**Resources:** Milad Yekani.

**Software:** Mohammad-Mahdi Abolghasemi, Shahriar Rezghi Shirsavar.

**Supervision:** Milad Yekani.

**Validation:** Milad Yekani.

**Visualization:** Mohammad-Mahdi Abolghasemi.

**Writing – original draft:** Mohammad-Mahdi Abolghasemi, Amirreza Bahramani, Milad Yekani.

**Writing – review & editing:** Mohammad-Mahdi Abolghasemi, Amirreza Bahramani, Milad Yekani.

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
