## [Decision Letter · Decision Letter 0]

2 Aug 2024

Dear Dr. yekani,

Thank you for submitting your manuscript to PLOS ONE. After careful consideration, we feel that it has merit but does not fully meet PLOS ONE’s publication criteria as it currently stands. Therefore, we invite you to submit a revised version of the manuscript that addresses the points raised during the review process.

In particular, the reviewers raised two substantive issues that must be addressed before we can consider your manuscript for publication.  First, for a new protocol to be demonstrated as an valid measure, it requires documentation that it actually measures EEG accurately; therefore, the technique needs to be compared to some type of a "gold standard".  Both reviewers raised this issue and provided excellent suggestions for comparison to existing techniques.  Ideally, a side-by-side comparison would be utilized but I will consider any reasonable option for validation of the approach.  Second, the writing for the protocol requires substantive revisions, as detailed by the reviewers.  Please address all reviewer suggestions prior to resubmitting for re-evaluation.

We look forward to receiving your revised manuscript.

Kind regards,

Brenton G. Cooper, Ph.D.

Academic Editor

PLOS ONE

Journal Requirements:

2. We note you have not yet provided a protocols.io PDF version of your protocol and/or a protocols.io DOI. When you submit your revision, please provide a PDF version of your protocol as generated by protocols.io (the file will have the protocols.io logo in the upper right corner of the first page) as a Supporting Information file. The filename should be S1_file.pdf, and you should enter “S1 File” into the Description field. Any additional protocols should be numbered S2, S3, and so on. Please also follow the instructions for Supporting Information captions [https://journals.plos.org/plosone/s/supporting-information#loc-captions]. The title in the caption should read: “Step-by-step protocol, also available on protocols.io.”

Please assign your protocol a protocols.io DOI, if you have not already done so, and include the following line in the Materials and Methods section of your manuscript: “The protocol described in this peer-reviewed article is published on protocols.io (https://dx.doi.org/10.17504/protocols.io.[...]) and is included for printing purposes as S1 File.” You should also supply the DOI in the Protocols.io DOI field of the submission form when you submit your revision.

If you have not yet uploaded your protocol to protocols.io, you are invited to use the platform’s protocol entry service [https://www.protocols.io/we-enter-protocols] for doing so, at no charge. Through this service, the team at protocols.io will enter your protocol for you and format it in a way that takes advantage of the platform’s features. When submitting your protocol to the protocol entry service please include the customer code PLOS2022 in the Note field and indicate that your protocol is associated with a PLOS ONE Lab Protocol Submission. You should also include the title and manuscript number of your PLOS ONE submission.

3. To comply with PLOS ONE submissions requirements, in your Methods section, please provide additional information regarding the experiments involving animals and ensure you have included details on (1) methods of sacrifice, (2) methods of anesthesia and/or analgesia, and (3) efforts to alleviate suffering.

Reviewers' comments:

Reviewer's Responses to Questions

**Comments to the Author**



Reviewer #1: Yes

Reviewer #2: Yes

2. Has the protocol been described in sufficient detail?

To answer this question, please click the link to protocols.io in the Materials and Methods section of the manuscript (if a link has been provided) or consult the step-by-step protocol in the Supporting Information files.

Reviewer #1: No

Reviewer #2: Yes

3. Does the protocol describe a validated method?

Reviewer #1: Yes

Reviewer #2: No

4. If the manuscript contains new data, have the authors made this data fully available?

Reviewer #1: N/A

Reviewer #2: Yes

**5. Is the article presented in an intelligible fashion and written in standard English?**

Reviewer #1: **No: ** Please see comments in main review: the manuscript is generally well written, but the protocols.io description has many grammatical and typographical errors, as well as corrupted formatting in the version that I accessed:

a few examples:

1. Punctuation, including periods at the end of sentences and capitalization, is often incorrect or missing.

e.g. "For delivering the external signal to the open bci board input pins on the board can be used. however, the external trigger signal source should be isolated to avoid electrical noise contamination of the signal this is done by an optocoupler circuit.”

2. Place the noise canceling and the grand electrode on the surface of the cerebellum (should be "ground electrode" here and elsewhere)

3. "ignalsEEG s during deep anesthesia, burst, and suppression periods are visually detectable" garbled text.

4. "open the open bci GUI, close the acoustic chamber, and then start the stimuli-presenting program. monitor the triggers arrivals on the GUI. We used pin 17 as inputs in the board if the". text is cutoff at this point in my version.

5. One step is duplicated (CHECK - 22 and 23?)

please proofread carefully and correct.

Reviewer #2: Yes

Reviewer #1: please see attached review.

What do you mean "minimum character count not met?

Reviewer #2: In this manuscript, the authors explain a method that can be used for EEG recordings in zebra finches. They show the presence of signals and then test the use of these signals by recording in two conditions, namely, (1) in anesthetised birds, they show that the duration of suppression of the signal is positively correlated with the 3 depths of anesthesia and (2) in awake birds, they record responses to 3 different auditory stimuli (3000 Hz tone, white noise and conspecific song) and show some differential responses.

Overall, this is a manuscript that describes a relatively inexpensive method for recording EEG signals from zebra finches. Though the usefulness of this method for further studies is not completely clear to me. I have outlined my concerns and suggestions below.

• The increase in duration of suppression with higher doses of anesthesia is robust. A few concerns and suggestions are listed here

◦ The authors should briefly explain how they calculated the duration of suppression. Currently they cite an existing paper and say that they followed this method. They should still briefly describe the calculation while including the citation. This allows the reader to get a sense of the method without having to go to the previous citation. The reader can go to the citation if they want more details

◦ Currently statistics are done in a pairwise fashion. This involves multiple comparisons and the p-value should be adjusted for multiple comparisons (https://en.wikipedia.org/wiki/Multiple_comparisons_problem#Controlling_procedures). It would be best to adjust p-values using the Bonferroni correction.

◦ The authors should also confirm that the bird's responses were also related to the level of anesthesia since there can be some inter-individual variability in response to anesthesia. Therefore, classifying the level based on concentration of isofluorane alone may not be enough.

• There is no quantification of differences in EEG responses to auditory stimuli. This is only qualitatively described. It is ok if this is just to show that the method does work. However, it would be useful to know where the electrodes were exactly. Given that this is iEEG responses, which I am assuming is intra-cranial EEG from the surface of the brain and it would be useful to know where these electrodes are? This would help understand what kinds of responses to expect.

• The authors need to compare their recordings with other methods for recording EEG (or recordings from other papers) to validate their method. In other words, are they really recording EEG signals and are their recordings similar to what has been described in the literature.

• An important aspect of the discussion is for the authors to explain how this technique could be useful and this is not explored. This could also be useful for characterizing sleep states.

• The authors do not define what "iEEG" is - I assumed it is intracranial EEG, but the authors should define this the first time they use the term. Is it not possible to record this from the skull?

• Why is this method superior to regular EEG recordings from the skull - possibly from on top of the second layer of skull? Again the introduction should really provide more information on the limitations of current methods and why this method is likely to be useful.

• Another useful addition would be to provide a table with costs for the entire system and compare this with costs for commercially available systems - this would support the claim that this is an affordable solution.

• Typo - "ground" electrodes are referred to as "grand" electrodes in the protocol.

**Do you want your identity to be public for this peer review?** For information about this choice, including consent withdrawal, please see our Privacy Policy

Reviewer #1: No

Reviewer #2: No

---

## [Author Response · Author response to Decision Letter 1]

12 Nov 2024

Response to the reviewers

We would like to express our sincere gratitude for your thorough review and insightful comments. Your feedback has significantly enhanced the quality and clarity of our work. We carefully considered each suggestion and integrated them into the revised version. We genuinely believe these improvements have strengthened our research, and we hope you will feel the same upon reviewing the updated manuscript.

In the following sections, we provide detailed responses to your comments, outlining the revisions and how they address the issues you raised. We are grateful for the opportunity to improve our manuscript through this process and look forward to your feedback on the revised version.

Sincerely,

Milad Yekani, DVM, PhD

Postdoctoral Researcher

Institute for Research in Fundamental Sciences (IPM)

miladyekani@ipm.ir

+98 914 347 8916

1. For a paper that is primarily intended to provide a protocol for the use of OpenBCI resources for iEEG recordings, you should provide significantly more detail within the paper and/or at protocols.io on how to get the system working, the specific modifications for use in the zebra finch, and any issues that have arisen with respect to troubleshooting or analysis. As much as possible, you should make this a resource for someone to assess whether this is the right technology for them to use, and to glean most of the information they need to implement what you have done. This includes:

a) perhaps in introduction and/or in the methods (by Methods here and in the rest of the review I am referring both to the methods section and to the protocols.io description), please provide a general description of Open BCI and some relevant citations or references to where one can go online to procure the relevant hardware and software to reproduce the authors’ system. The paper itself lists reference 24 as having some of this information, but no reference 24 is included in the bibliography. More generally, at several points the authors say they followed procedures as described in some other reference. It is fine to direct people to a reference for further details, but please at least describe enough within the methods of this paper itself so that readers have an understanding of the relevant details. If Open BCI has different versions of hardware and/or software, please make note of this and detail which were used in this study; if there are choices here for different applications, perhaps include a table or further description that both specifies what was used here and what other options might be available.

These additions are to fully orient readers to the general resources that were already available that you have used in the study and that readers could access.

The comments mentioned above have been addressed as explained below:

General description of the OpenBCI:

We have added a description of OpenBCI. The OpenBCI Cyton Biosensing Board was used for this study, and we have provided relevant links for obtaining the hardware and software directly from the OpenBCI website. We have also described alternative hardware configurations, such as the Ganglion board (Protocol.io, Section 1, "Getting Started with OpenBCI"). We have also included a general description of OpenBCI in the introduction and provided related references. This introductory part mainly focuses on OpenBCI’s capabilities and hardware features.

Please refer to the fifth paragraph of the article's introduction: “OpenBCI offers a versatile open-source hardware platform for various biosensing applications, including EEG, EMG, and ECG signals. At its core, the OpenBCI Cyton board integrates several key components: a PIC32MX250F128B microcontroller, a ChipKIT UDB32-MX2-DIP bootloader, an ADS1299 analog-to-digital converter with eight channels (expandable to 16) and a maximum sampling rate of 16 kHz, and a 3-axis LIS3DH accelerometer. Users can configure EEG channels in either monopolar or bipolar modes, while the board also accommodates five external digital inputs and three analog inputs. Moreover, the device supports data streaming over Wi-Fi via Transmission Control Protocol (TCP)”.

Step by step instruction for OpenBCI:

To make it easier for readers to reproduce our setup, we have included step-by-step instructions in the protocol.io description on assembling and configuring the OpenBCI system for recording iEEG data. (Section 2, Assembling the Recording Apparatus).

b) Some aspects of the study will be specific to your modifications for use in songbirds, and this should also be more clearly spelled out so that others do not have to reinvent what you have done. For example, please specify in more detail how the electrodes were constructed (perhaps with photo) and perhaps anything further that you can note about whether or not the properties of construction matter. Were these electrodes that were provided by Open BCI or another commercial source, or did you construct them? What was the material, what was the tip exposure like, what was the impedance? Protocols.io is a good place for much of this, so please make sure to direct the reader clearly to that resource for information. It would be very helpful to specify some further detail, perhaps showing more specifically a parts list of materials acquired from OpenBCI, and additional materials that you needed to add to run these experiments, and this could be worth putting in a table or figure.

The comments mentioned above have been addressed as explained below:

Electrode preparation: The new protocol describes how to prepare the electrodes. It includes steps for bending, shaping, and disinfecting the electrode before usage. Regarding the concerns about disinfection by ethanol, we have reviewed all procedures with our veterinarian. We were advised to avoid contact and ensure all ethanol was evaporated and rinsed with saline. We also reminded readers to consult their veterinarian, as these methods may not agree with their institute's regulations and protocols. See Protocol.io, Section 4, "Electrode Preparation".

Electrode placement: The protocol explains how electrodes are placed under the skull and on the dura mater and how force is applied to ensure they are correctly fixed. It also describes how electrodes are pushed into position, including a diagram showing electrode placement in relation to the skull and brain. See Protocol.io, Section 28, "Electrode Placement and Fixation."

Parts list and materials: A detailed list of materials includes stainless steel wire, hypodermic needles, dental cement, and other supplies needed to construct and implant the electrodes. It also specifies where to purchase these materials and necessary tools. See Protocol.io, "Materials".

Electrode montage and recording setup: This section outlines how electrodes are connected to the OpenBCI board and details the positions (e.g., cerebellum for reference electrodes). See Protocol.io, Section 7

Electrode impedance: In protocol.io, we have provided a comprehensive explanation of how the electrode's impedance was measured and its range (15k-20k). See Protocol.io, Section 8.

c) How many electrodes were placed in or on the brain, where were they placed, and how much does it matter? I could find no detail either on the general placement electrodes, or how they contacted the brain. For example, were these resting on the meninges? Or were the meninges removed, or were these sitting on the inner layer of skull, etc? Was any kind of conductive material used to enhance conductivity and signal-to-noise ratio? How are they affixed to the brain? At one point it was mentioned that there is a cerebellar reference electrode. Where and how was that placed, and how were the signals from each of the other electrodes connected to the amplifiers/hardware provided by Open BCI?

We have carefully addressed this comment and have added further details about the placement and contact of the electrodes in the protocol.io file. Specifically:

Number of electrodes: We clarified that four electrodes were used in our setup: two for signal recordings and two for reference and noise cancellation. These electrodes were positioned in both frontal and auditory regions, and the reference electrodes were located on the cerebellum. These placements have been specified in greater detail, including how their locations were determined using bifurcation (Protocol.io, Section 6, "Electrode Montage").

Electrode contact with the brain: We explicitly stated that the electrodes were placed under the skull and above the meninges (i.e., resting on the dura mater). We have added this clarification in the revised protocol (Section 26, "Electrode Placement and Fixation"). The meninges were not removed during the process, and no conductive material was used to enhance signal conductivity, as we determined the natural contact between the electrode and dura mater was sufficient for our recordings. Speak about the impedance of the electrodes.

Affixing the electrodes to the brain: The protocol now includes a detailed description of how electrodes were fixed in place. The force applied between the skull and brain was used to maintain stable positioning of the electrodes (Section 27, "Electrode Placement and Fixation"). This is illustrated in the updated diagram, which shows how the electrode was positioned and maintained under the skull until we fixed it permanently. Eventually, it can be firmly fixed by cementing the body of the electrode to the skull.

Reference Electrode: We have elaborated on the placement of the reference electrodes on the cerebellum. These electrodes were placed at separate cerebellar locations and connected to the noise cancellation system. We also specified how these electrodes were connected to the OpenBCI board for optimal signal recording (Section 6, "Electrode Montage"). All signal electrodes were referenced to a ground electrode placed on the cerebellum, and no differential signals were compared between pairs of recording electrodes (Section 6, "Electrode Montage").

Signal differentiation: We clarified that the signal was differentiated relative to the ground electrode on the cerebellum. The signal electrodes recorded activity from the brain regions of interest (e.g., frontal lobe, auditory cortex), and these signals were compared relative to the cerebellar ground electrode, not across pairs of recording electrodes.

d) With respect to electrode placement, please clarify how the tips of the electrodes were positioned relative to relevant landmarks such as the bifurcation of the mid-sagittal sinus. Were all electrodes referenced to the cerebellum, or were there pairs of electrodes across which differential signals were Compared?

As described in the revised manuscript and on protocols.io, the first electrode was positioned over the HVC (0.2 mm AP, 2.3 mm ML relative to the bifurcation), while the second electrode was positioned over the CMM (2 mm AP, 0.4 mm ML relative to the bifurcation). All signal electrodes were referenced to a ground electrode placed on the cerebellum, and no differential signals were compared between pairs of recording electrodes (Section 6, "Electrode Montage").

e) with respect to electrode placement, can the authors provide any guidance on how much it matters? For EEG, usually, there is quite a bit of dependency of signals on electrode placement. The authors show data from “anterior” and “posterior” electrodes, but do not specify the coordinates where these were placed. We can see that there are some differences between the anterior and the poster electrodes in terms of the signals that are recorded. However, it would be helpful if the authors could provide some more sense of the spatial variation in signals that would help readers understand what kind of spatial resolution is there for distinguishing signals between different brain areas: The authors describe one of the advantages of iEEG as potentially providing a greater “survey“ of activity distribution across the brain than is normally achieved with electrophysiology. This would be substantiated and clarified to the extent that signals from nearby recording locations look different from each other.

In the new experiments, we placed the electrodes precisely on two important superficial nuclei, HVC and CMM. The exact coordinates of these electrodes are detailed in the protocol and the revised manuscript (Section 24, "Localizing the electrode sites"). Regarding signal localization in our new design, we found some differences between the signals of different electrodes, which have been explained in the result section under the Auditory Response Recording part.” Interestingly, the ERP response for the BOS differs significantly from the other stimuli; it exhibits a later rise with a smaller slope. This may be due to the distinct responses of these nuclei to the BOS. These differences in response were also distinctly observed in the HVC and CMM. A larger amplitude in the CMM compared to the HVC may result from the dominant role of the CMM in processing auditory stimuli, as is evident across all conditions (Figure 3A, B).”

We have also mentioned that modifying electrodes can strengthen localization power (see discussion in the article). Although we observed notable differences between the responses to BOS and reverse BOS in the HVC and CMM electrodes, we should consider that localization can be improved by using higher-impedance electrodes and limiting the contact area. These modifications could also allow us to implant electrodes directly within the nuclei, enabling us to capture LFP signals with greater spatial resolution.

f) It would be useful to sketch out more clearly any additional workflow that the authors can provide, for example with respect to how the signals were recorded, and what software was used to process them, etc., so that readers could take advantage of the protocol without needing to reinvent various components or go to other sources. If these components are all provided within Open BCI, that should be clarified with appropriate references.

Signal Recording: The OpenBCI Cyton board was used to acquire signals, and recordings were managed through the OpenBCI GUI. Sections 5, 7, and 13 provide details on setup, electrode connections, and troubleshooting.

Software and Analysis: Data was processed in Python using the MNE library for filtering, epoch creation, and ERP analysis. Section 15 provides step-by-step processing details and references.

References and Guides: We clarified the use of OpenBCI components and provided direct links to guides and troubleshooting resources. For complete documentation, see Sections 3, 5, and 13.

2) For the specifics of experimental results that are shown in figures, please also consider where more detail might be helpful for a reader. This again includes some aspects of methods that should be in this paper and not included by reference. For example, for suppression duration shown in figure 2, how was this measured? For low medium, and high levels of isofluorane, how long were animals at a fixed level of anesthesia before measurements were made? For figure 3, is this the average of some number of stimulus presentations or a single trial? What was the stimulus duration in these figures? The stimulus duration is particularly important to show, because of the structure of responses that show, for example, some peak at 1.1 seconds - does that relate in any way to when the stimulus stopped or changed in amplitude? Showing oscillograms of the stimuli here would be helpful and could also address one other question, which is how loud were the stimuli?

Thank you very much for your accurate comments. More information has been added to the paper overall to provide more detail and value for the reader. Below are examples of information added as an answer to the specific points mentioned.

Suppression Duration: We clarified how suppression duration was measured by following the method of Westover et al. (2013), with a defined threshold to segment suppressions and bursts. Details on the stabilization period: Animals were kept at each isoflurane level for 2 minutes. As reaching the steady state of anesthesia occurs immediately, we used the signal immediately after changing the concentrati

---

## [Editor Report · Decision Letter 1]

13 Dec 2024

Dear Dr. yekani,

Thank you for submitting your manuscript to PLOS ONE. After reviewing your manuscript and consulting with one of the two reviewers, we were unable to find your response and changes to the manuscript to Reviewer #2's comments.  Below you will find the reviewer's comments.  We invite you to submit a revised version of the manuscript that addresses the reviewer's points and at that time, I will send your manuscript back out for review.

We look forward to receiving your revised manuscript.

Kind regards,

Brenton G. Cooper, Ph.D.

Academic Editor

PLOS ONE

Reviewers' comments:

Reviewer #2: In this manuscript, the authors explain a method that can be used for EEG recordings in zebra finches. They show the presence of signals and then test the use of these signals by recording in two conditions, namely, (1) in anesthetised birds, they show that the duration of suppression of the signal is positively correlated with the 3 depths of anesthesia and (2) in awake birds, they record responses to 3 different auditory stimuli (3000 Hz tone, white noise and conspecific song) and show some differential responses.

Overall, this is a manuscript that describes a relatively inexpensive method for recording EEG signals from zebra finches. Though the usefulness of this method for further studies is not completely clear to me. I have outlined my concerns and suggestions below.

• The increase in duration of suppression with higher doses of anesthesia is robust. A few concerns and suggestions are listed here

◦ The authors should briefly explain how they calculated the duration of suppression. Currently they cite an existing paper and say that they followed this method. They should still briefly describe the calculation while including the citation. This allows the reader to get a sense of the method without having to go to the previous citation. The reader can go to the citation if they want more details

◦ Currently statistics are done in a pairwise fashion. This involves multiple comparisons and the p-value should be adjusted for multiple comparisons (https://en.wikipedia.org/wiki/Multiple_comparisons_problem#Controlling_procedures). It would be best to adjust p-values using the Bonferroni correction.

◦ The authors should also confirm that the bird's responses were also related to the level of anesthesia since there can be some inter-individual variability in response to anesthesia. Therefore, classifying the level based on concentration of isofluorane alone may not be enough.

• There is no quantification of differences in EEG responses to auditory stimuli. This is only qualitatively described. It is ok if this is just to show that the method does work. However, it would be useful to know where the electrodes were exactly. Given that this is iEEG responses, which I am assuming is intra-cranial EEG from the surface of the brain and it would be useful to know where these electrodes are? This would help understand what kinds of responses to expect.

• The authors need to compare their recordings with other methods for recording EEG (or recordings from other papers) to validate their method. In other words, are they really recording EEG signals and are their recordings similar to what has been described in the literature.

• An important aspect of the discussion is for the authors to explain how this technique could be useful and this is not explored. This could also be useful for characterizing sleep states.

• The authors do not define what "iEEG" is - I assumed it is intracranial EEG, but the authors should define this the first time they use the term. Is it not possible to record this from the skull?

• Why is this method superior to regular EEG recordings from the skull - possibly from on top of the second layer of skull? Again the introduction should really provide more information on the limitations of current methods and why this method is likely to be useful.

• Another useful addition would be to provide a table with costs for the entire system and compare this with costs for commercially available systems - this would support the claim that this is an affordable solution.

• Typo - "ground" electrodes are referred to as "grand" electrodes in the protocol.

---

## [Author Response · Author response to Decision Letter 2]

27 Jan 2025

Responses to reviewer 1’s comments

1. For a paper that is primarily intended to provide a protocol for the use of OpenBCI resources for iEEG recordings, you should provide significantly more detail within the paper and/or at protocols.io on how to get the system working, the specific modifications for use in the zebra finch, and any issues that have arisen with respect to troubleshooting or analysis. As much as possible, you should make this a resource for someone to assess whether this is the right technology for them to use, and to glean most of the information they need to implement what you have done. This includes:

a) perhaps in introduction and/or in the methods (by Methods here and in the rest of the review I am referring both to the methods section and to the protocols.io description), please provide a general description of Open BCI and some relevant citations or references to where one can go online to procure the relevant hardware and software to reproduce the authors’ system. The paper itself lists reference 24 as having some of this information, but no reference 24 is included in the bibliography. More generally, at several points the authors say they followed procedures as described in some other reference. It is fine to direct people to a reference for further details, but please at least describe enough within the methods of this paper itself so that readers have an understanding of the relevant details. If Open BCI has different versions of hardware and/or software, please make note of this and detail which were used in this study; if there are choices here for different applications, perhaps include a table or further description that both specifies what was used here and what other options might be available.

These additions are to fully orient readers to the general resources that were already available that you have used in the study and that readers could access.

Response:

General description of the OpenBCI:

We have added a description of OpenBCI. The OpenBCI Cyton Biosensing Board was used for this study, and we have provided relevant links for obtaining the hardware and software directly from the OpenBCI website. We have also described alternative hardware configurations, such as the Ganglion board (Protocol.io, Section 1, "Getting Started with OpenBCI"). We have also included a general description of OpenBCI in the introduction and provided related references. This introductory part mainly focuses on OpenBCI’s capabilities and hardware features.

Please refer to the fifth paragraph of the article's introduction: “OpenBCI offers a versatile open-source hardware platform for various biosensing applications, including EEG, EMG, and ECG signals. At its core, the OpenBCI Cyton board integrates several key components: a PIC32MX250F128B microcontroller, a ChipKIT UDB32-MX2-DIP bootloader, an ADS1299 analog-to-digital converter with eight channels (expandable to 16) and a maximum sampling rate of 16 kHz, and a 3-axis LIS3DH accelerometer. Users can configure EEG channels in either monopolar or bipolar modes, while the board also accommodates five external digital inputs and three analog inputs. Moreover, the device supports data streaming over Wi-Fi via Transmission Control Protocol (TCP)”.

Step by step instruction for OpenBCI:

To make it easier for readers to reproduce our setup, we have included step-by-step instructions in the protocol.io description on assembling and configuring the OpenBCI system for recording iEEG data. (Section 2, Assembling the Recording Apparatus).

b) Some aspects of the study will be specific to your modifications for use in songbirds, and this should also be more clearly spelled out so that others do not have to reinvent what you have done. For example, please specify in more detail how the electrodes were constructed (perhaps with photo) and perhaps anything further that you can note about whether or not the properties of construction matter. Were these electrodes that were provided by Open BCI or another commercial source, or did you construct them? What was the material, what was the tip exposure like, what was the impedance? Protocols.io is a good place for much of this, so please make sure to direct the reader clearly to that resource for information. It would be very helpful to specify some further detail, perhaps showing more specifically a parts list of materials acquired from OpenBCI, and additional materials that you needed to add to run these experiments, and this could be worth putting in a table or figure.

Response:

Electrode preparation: The new protocol describes how to prepare the electrodes. It includes steps for bending, shaping, and disinfecting the electrode before usage. Regarding the concerns about disinfection by ethanol, we have reviewed all procedures with our veterinarian. We were advised to avoid contact and ensure all ethanol was evaporated and rinsed with saline. We also reminded readers to consult their veterinarian, as these methods may not agree with their institute's regulations and protocols. See Protocol.io, Section 4, "Electrode Preparation".

Electrode placement: The protocol explains how electrodes are placed under the skull and on the dura mater and how force is applied to ensure they are correctly fixed. It also describes how electrodes are pushed into position, including a diagram showing electrode placement in relation to the skull and brain. See Protocol.io, Section 28, "Electrode Placement and Fixation."

Parts list and materials: A detailed list of materials includes stainless steel wire, hypodermic needles, dental cement, and other supplies needed to construct and implant the electrodes. It also specifies where to purchase these materials and necessary tools. See Protocol.io, "Materials".

Electrode montage and recording setup: This section outlines how electrodes are connected to the OpenBCI board and details the positions (e.g., cerebellum for reference electrodes). See Protocol.io, Section 7

Electrode impedance: In protocol.io, we have provided a comprehensive explanation of how the electrode's impedance was measured and its range (15k-20k). See Protocol.io, Section 8.

c) How many electrodes were placed in or on the brain, where were they placed, and how much does it matter? I could find no detail either on the general placement electrodes, or how they contacted the brain. For example, were these resting on the meninges? Or were the meninges removed, or were these sitting on the inner layer of skull, etc? Was any kind of conductive material used to enhance conductivity and signal-to-noise ratio? How are they affixed to the brain? At one point it was mentioned that there is a cerebellar reference electrode. Where and how was that placed, and how were the signals from each of the other electrodes connected to the amplifiers/hardware provided by Open BCI?

Response:

We have carefully addressed this comment and have added further details about the placement and contact of the electrodes in the protocol.io file. Specifically:

Number of electrodes: We clarified that four electrodes were used in our setup: two for signal recordings and two for reference and noise cancellation. These electrodes were positioned in both frontal and auditory regions, and the reference electrodes were located on the cerebellum. These placements have been specified in greater detail, including how their locations were determined using bifurcation (Protocol.io, Section 6, "Electrode Montage").

Electrode contact with the brain: We explicitly stated that the electrodes were placed under the skull and above the meninges (i.e., resting on the dura mater). We have added this clarification in the revised protocol (Section 26, "Electrode Placement and Fixation"). The meninges were not removed during the process, and no conductive material was used to enhance signal conductivity, as we determined the natural contact between the electrode and dura mater was sufficient for our recordings. Speak about the impedance of the electrodes.

Affixing the electrodes to the brain: The protocol now includes a detailed description of how electrodes were fixed in place. The force applied between the skull and brain was used to maintain stable positioning of the electrodes (Section 27, "Electrode Placement and Fixation"). This is illustrated in the updated diagram, which shows how the electrode was positioned and maintained under the skull until we fixed it permanently. Eventually, it can be firmly fixed by cementing the body of the electrode to the skull.

Reference Electrode: We have elaborated on the placement of the reference electrodes on the cerebellum. These electrodes were placed at separate cerebellar locations and connected to the noise cancellation system. We also specified how these electrodes were connected to the OpenBCI board for optimal signal recording (Section 6, "Electrode Montage"). All signal electrodes were referenced to a ground electrode placed on the cerebellum, and no differential signals were compared between pairs of recording electrodes (Section 6, "Electrode Montage").

Signal differentiation: We clarified that the signal was differentiated relative to the ground electrode on the cerebellum. The signal electrodes recorded activity from the brain regions of interest (e.g., frontal lobe, auditory cortex), and these signals were compared relative to the cerebellar ground electrode, not across pairs of recording electrodes.

d) With respect to electrode placement, please clarify how the tips of the electrodes were positioned relative to relevant landmarks such as the bifurcation of the mid-sagittal sinus. Were all electrodes referenced to the cerebellum, or were there pairs of electrodes across which differential signals were Compared?

Response:

As described in the revised manuscript and on protocols.io, the first electrode was positioned over the HVC (0.2 mm AP, 2.3 mm ML relative to the bifurcation), while the second electrode was positioned over the CMM (2 mm AP, 0.4 mm ML relative to the bifurcation). All signal electrodes were referenced to a ground electrode placed on the cerebellum, and no differential signals were compared between pairs of recording electrodes (Section 6, "Electrode Montage").

e) with respect to electrode placement, can the authors provide any guidance on how much it matters? For EEG, usually, there is quite a bit of dependency of signals on electrode placement. The authors show data from “anterior” and “posterior” electrodes, but do not specify the coordinates where these were placed. We can see that there are some differences between the anterior and the poster electrodes in terms of the signals that are recorded. However, it would be helpful if the authors could provide some more sense of the spatial variation in signals that would help readers understand what kind of spatial resolution is there for distinguishing signals between different brain areas: The authors describe one of the advantages of iEEG as potentially providing a greater “survey“ of activity distribution across the brain than is normally achieved with electrophysiology. This would be substantiated and clarified to the extent that signals from nearby recording locations look different from each other.

Response:

In the new experiments, we placed the electrodes precisely on two important superficial nuclei, HVC and CMM. The exact coordinates of these electrodes are detailed in the protocol and the revised manuscript (Section 24, "Localizing the electrode sites"). Regarding signal localization in our new design, we found some differences between the signals of different electrodes, which have been explained in the result section under the Auditory Response Recording part.” Interestingly, the ERP response for the BOS differs significantly from the other stimuli; it exhibits a later rise with a smaller slope. This may be due to the distinct responses of these nuclei to the BOS. These differences in response were also distinctly observed in the HVC and CMM. A larger amplitude in the CMM compared to the HVC may result from the dominant role of the CMM in processing auditory stimuli, as is evident across all conditions (Figure 3A, B).”

We have also mentioned that modifying electrodes can strengthen localization power (see discussion in the article). Although we observed notable differences between the responses to BOS and reverse BOS in the HVC and CMM electrodes, we should consider that localization can be improved by using higher-impedance electrodes and limiting the contact area. These modifications could also allow us to implant electrodes directly within the nuclei, enabling us to capture LFP signals with greater spatial resolution.

f) It would be useful to sketch out more clearly any additional workflow that the authors can provide, for example with respect to how the signals were recorded, and what software was used to process them, etc., so that readers could take advantage of the protocol without needing to reinvent various components or go to other sources. If these components are all provided within Open BCI, that should be clarified with appropriate references.

Response:

Signal Recording: The OpenBCI Cyton board was used to acquire signals, and recordings were managed through the OpenBCI GUI. Sections 5, 7, and 13 provide details on setup, electrode connections, and troubleshooting.

Software and Analysis: Data was processed in Python using the MNE library for filtering, epoch creation, and ERP analysis. Section 15 provides step-by-step processing details and references.

References and Guides: We clarified the use of OpenBCI components and provided direct links to guides and troubleshooting resources. For complete documentation, see Sections 3, 5, and 13.

2) For the specifics of experimental results that are shown in figures, please also consider where more detail might be helpful for a reader. This again includes some aspects of methods that should be in this paper and not included by reference. For example, for suppression duration shown in figure 2, how was this measured? For low medium, and high levels of isofluorane, how long were animals at a fixed level of anesthesia before measurements were made? For figure 3, is this the average of some number of stimulus presentations or a single trial? What was the stimulus duration in these figures? The stimulus duration is particularly important to show, because of the structure of responses that show, for example, some peak at 1.1 seconds - does that relate in any way to when the stimulus stopped or changed in amplitude? Showing oscillograms of the stimuli here would be helpful and could also address one other question, which is how loud were the stimuli?

Response:

Thank you very much for your accurate comments. More information has been added to the paper overall to provide more detail and value for the reader. Below are examples of information added as an answer to the specific points mentioned.

Suppression Duration: We clarified how suppression duration was measured by following the method of Westover et al. (2013), with a defined threshold to segment suppressions and bursts. Details on the stabilization period: Animals were kept at each isoflurane level for 2 minutes. As reaching the steady state of anesthesia occurs immediately, we used the signal immediately after changing the concentration (Protocol, section 9).

Stimulus Presentations: we have provided a detailed description of stimuli (duration, number of repetitions, loudness, spectrogram) (protocol section 11). In the article's discussion section, we have also discussed the features of brain responses and their possible relationships to the stimuli.

3) as noted at the top, the more that you can show that this set up records useful signals that are similar to those from other eeg recordings in birds, the more likely readers will follow your approach. In that respect, please look a little bit more for songbird references that have recorded EEG’s. Low and Margoliash published a paper (“Mammalian-like features of sleep structure in zebra finches “ PNAS 2008) that I tracked down, but I believe there may be others, including from Low. It would be worth citing this paper and comparing your approach with theirs; if yo

---

## [Decision Letter · Decision Letter 2]

1 Apr 2025

An affordable solution for investigating zebra finch intracranial electroencephalography (iEEG) signals

PONE-D-24-21612R2

Dear Dr. yekani,

We’re pleased to inform you that your manuscript has been judged scientifically suitable for publication and will be formally accepted for publication once it meets all outstanding technical requirements.

Kind regards,

Brenton G. Cooper, Ph.D.

Academic Editor

PLOS ONE

Reviewers' comments:

Reviewer's Responses to Questions

**Comments to the Author**



Reviewer #2: Yes

2. Has the protocol been described in sufficient detail?

To answer this question, please click the link to protocols.io in the Materials and Methods section of the manuscript (if a link has been provided) or consult the step-by-step protocol in the Supporting Information files.

Reviewer #2: Yes

3. Does the protocol describe a validated method?

Reviewer #2: Yes

4. If the manuscript contains new data, have the authors made this data fully available?

Reviewer #2: Yes

**5. Is the article presented in an intelligible fashion and written in standard English?**

Reviewer #2: Yes

Reviewer #2: The authors have addressed all of my comments appropriately. This study can now be published in PLOS ONE.

**Do you want your identity to be public for this peer review?** For information about this choice, including consent withdrawal, please see our Privacy Policy

Reviewer #2: No

---

## [Editor Report · Acceptance letter]

PONE-D-24-21612R2

PLOS ONE

Dear Dr. yekani,

I'm pleased to inform you that your manuscript has been deemed suitable for publication in PLOS ONE. Congratulations! Your manuscript is now being handed over to our production team.

Kind regards,

on behalf of

Dr. Brenton G. Cooper

Academic Editor

PLOS ONE